# Financial Literacy: The Case of Poland

**Beata Swiecka [1], Eser Yeşildağ [2], Ercan Özen [2] and Simon Grima [3,*]** 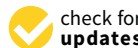

[1]  Institute of Economics and Finance, University of Szczecin, 71-004 Szczecin, Poland;
    beata.swiecka@usz.edu.pl
[2]  School of Applied Sciences, The University of Usak, Usak 64200, Turkey; eser.yesildag@usak.edu.tr (E.Y.);
    ercan.ozen@usak.edu.tr (E.Ö.)
[3]  Faculty of Economics, Management & Accountancy, The University of Malta, Msida, MSD 2080, Malta
*   Correspondence: simon.grima@um.edu.mt

**Abstract:** Financial literacy is a path to sustainability and has an important role in ensuring the financial sustainability of individuals, families, enterprises and national economies. The level of these economic indicators such as debt, payment discipline, savings and financial management all translate into prosperity or insolvency and bankruptcy and result partially from financial literacy. The higher the level of financial literacy, especially of young people, the more favourable the level of economic indicators, which translates into the economy and sustainable development. With this study we aim to determine the level of financial literacy of high school students in Poland and to determine whether financial literacy changes according to gender. The most important element that distinguishes our study from the others is that or study was carried out with a large sample of high school students with an average age of 15–16 years. In addition, the effect of gender on financial literacy at an early age was investigated, also comparing the wider themes to the so-called narrow themes. The results of the research demonstrated a good and partially very good, level of financial knowledge of the young people in Poland. 45.3% obtained an average level score and 43.8% achieved a high-level score in financial knowledge. This result shows that they can be rational in their financial decision making. However although, it is understood that gender makes a difference on financial behaviour and use of financial instruments, gender does not make any difference on the level of financial knowledge. Moreover, the financial literacy level of males is found to be higher than females.

**Keywords:** financial literacy; financial knowledge; financial attitude; financial behaviour; household finance; young people

## 1. Introduction

Financial literacy is important to ensure the sustainable development of individuals and society. According to Bryant, economic growth and sustainability are rooted in the financial literacy of individuals [1]. Rahmandoust show the importance of financial literacy in entrepreneurs' success and then in the sustainable development of society [2]. Financial literacy is important for the sustainability of both the consumers and the entrepreneurs and for people of all age. The earlier that financial literacy is acquired the greater the benefit for their development. Therefore, to ensure the sustainability of an economy, further studies are required to understand the impact it has. After the latest large financial crisis stating in 2008, The U.S. President's Advisory Council on Financial Literacy (PACFL 2008) [3] noted that:

> "far too many Americans do not have the basic financial skills necessary to develop and maintain a budget, to understand credit, to understand investment vehicles, or to take advantage of our banking system. It is essential to provide basic financial education that allows people to better navigate an economic crisis such as this one".

We understand that having enough financial knowledge and abilities resulted in appropriate and informed decisions, which are important not only for individuals and the local community but also on an international level. The Former U.S. Federal Reserve Board Chairman Ben Bernanke [4] informed that:

> "In our dynamic and complex financial marketplace, financial education must be a life-long pursuit that enables consumers of all ages and economic positions to stay attuned to changes in their financial needs and circumstances and to take advantage of products and services that best meet their goals. Well-informed consumers, who can serve as their own advocates, are one of the best lines of defence against the proliferation of financial products and services that are unsuitable, unnecessarily costly, or abusive."

In his speech Ben Bernanke stated that more financial literacy is important for a healthy financial and economic life. In addition, the impacts of Financial Literacy can be seen as spilling over into other different areas. For example, more literate people are more likely to participate in capital markets [5], meaning that more capital is collected by companies, which in, enables them to produce more, resulting in in more income, more employment and more wealth.

During the past decade, research on financial literacy has increased, but it is still defined heterogeneously by scientists, who indicate that there is further need to explore the subject. It is still being studied by scientists from around the world, which indicates the importance of this topic and the need for further exploration. Financial literacy is a multi-layered issue of great importance for the economy, society and sustainable development. Given the growing importance of sustainable development, as well as financial literacy, the authors decided to refer to financial literacy, which directly relates to the achievement of the objectives of sustainable development goals adopted by UN member states of all member states at the Sustainable Development Summit in New York in 2015 year. One of the goals refers to the issue of women, pointing to their enormous importance in the sustainable development of the world, which is why we decided to include gender issues in relation to the individual elements of financial literacy.

## 2. Literature Review

Financial literacy is a significant element in understanding finances and rational financial decision making. It influences the quality of financial life and the proper rational decision making in the area of finance [6]. Increasing financial literacy is vital importance for the public to improve welfare through better decision making [7]. Herein, the author addressed the financial literacy using themes such as knowledge, education, behaviour and well-being. Life's financial decisions depend on individuals' knowledge and understanding of personal finance [8]. The three fundamental financial decisions, important at various stages of one's financial life, include savings, making debts, and consumption. The quality of financial decisions taken into account by individuals depends on their financial knowledge, abilities, and attitudes [9]. Agarwalla et al., affirm that considering an environment where the scope and complexity of financial products and services keep increasing, it is necessary that the individuals develop a solid understanding on the word of finance to be able to become good decision-makers and use the right path to achieve their financial goals and needs [10].

Financial literacy is a broad concept, not explicitly defined by the authors. It can be defined as an ability to use knowledge and skills to manage one's financial resources effectively for lifetime financial security [11]. The dimension of financial literacy is a particular kind of human capital that is acquired throughout the life cycle, by learning various subjects that affect the ability to effectively manage revenue, expenses, and savings [12,13]. The Organization for Economic Co-operation Development International Network on Financial Education (OECD/INFE), defines financial literacy as a combination of awareness, knowledge, skill, attitude and behaviour necessary to make sound financial decisions and ultimately achieve individual financial wellbeing [14,15].

According to Orton, financial literacy can be divided into three dimensions; (i) financial knowledge and understanding, (ii) financial skills and competence, and (iii) financial responsibility [16]. Widdowson and Hailwood identify the following dimensions of financial literacy: (i) basic numeracy skills and basic arithmetic ability, (ii) understanding of the benefits and risks associated with particular financial decisions, and (iii) capability to know where to seek professional advice [14,17]. Remund groups financial literacy into five categories: (i) financial knowledge, knowledge about financial concepts and products; (ii) financial communication, communication aptitudes concerning financial concepts; (iii) financial ability, ability to use knowledge in order to take the necessary financial decisions; (iv) financial behaviour, real use of different financial instruments; (v) financial confidence people's confidence in their previous financial decisions and actions [18]. Atkinson and Messy define financial literacy as awareness, knowledge, skills, attitude, and behaviours [19]. The literature uses many different components of the concept of financial literacy. In their article resulting from primary research carried out with the participation of representatives involved in financial education and the promotion of financial knowledge, it was recognized that the Robson [20] approach adopted from Kempson [21], is one of the most appropriate, since it presents financial literacy as a collection of financial knowledge, financial behaviour and financial attitude [22]. Świecka et al. in addition to the above, also presents financial skills supplemented with financial information demand [23]. They suggested that financial literacy is broader and studies should include the concepts of financial knowledge, financial behaviour, financial attitude, financial skills and financial information demand. In this article three main components of financial literacy which have the greatest impact on the level of financial literacy, have been studied.

Various studies on financial literacy have defined this concept differently and studied it using different methods and frameworks. Although, the concept conceptualization adopted in the primary research referred to in this article, assumed that financial literacy consists of concepts such as: financial knowledge, financial skills, financial attitudes, financial behavioural, and financial information demand [23], in this article we have decided to include only and exclusively the three most important aspects as per studies by Kempson [22] and Robson [20]. This since a narrow approach to financial literacy allowed for the comparison to the broad concept of financial literacy. That is the 5 components: 1. financial knowledge, 2. financial skills, 3. financial attitudes, 4. financial behavioral 5. Financial information demand to the narrow view represented in this article which includes three elements: 1. financial knowledge, 2. financial attitudes, 3. financial behavioural.

The largest international survey of young people (15-year old students) is the Program for International Student Assessment (PISA). It has been implemented every 3 years since 2000 in all the OECD countries, as well as in dozens of partner countries. This financial literacy research was carried out for the first time in 2012. Poland has participated in the PISA survey from the beginning, i.e., from 2000. In 2012 the financial literacy study was conducted on a sample of 29,000 15-year-old students from 13 OECD countries. One of the aims of this study was to check whether the 15-year-olds have enough knowledge and skills to further their livelihood. In the PISA research conducted in 2015 [22] the Polish 15-years old students scored significantly poorer compared to the study carried out in 2012 (published in 2015). The PISA data (2017) indicates to what extent the 15-years students are already using money and are involved in financial decisions. On average, in 10 participating OECD countries about six out of ten students have a bank account and/or a prepaid card. More than half the students in Australia, the Flemish Community of Belgium, the Canadian provinces, Italy, the Netherlands, Spain, and the United States have a bank account and/or a prepaid card. Moreover, students also earn some money from small occasional jobs outside of school hours. In the PISA 2015 study [22], the Polish 15-year-olds scored 485 points as compared to the score of 510 points in PISA 2012. This was below the OECD average, which score was 489 points.

In 2015, the Chinese scored the most points, i.e., 566 as compared to their score of 603 in 2012—603 and the Belgians scored 541 in 2015 and 2012. In 2012, the USA, Russia, France, Slovenia, Spain, Croatia, Israel, Slovakia, Italy, and Colombia scored lower results than Poland [24]. However, in the

next survey, conducted in 2015, only Italy, Spain, Lithuania, Slovak Republic, Chile, Peru, and Brazil had lower results than Poland and with Russia with 512 points and the USA with 587 points exceeding Poland's score [25].

There are different factors that affect financial literacy. One of them is gender. In most studies reported in the literature, males have a higher level of financial literacy than females. In the studies on households [13], [26–30] and the studies on students [31–33], males exhibited higher level of financial literacy than females. Some studies on university students [34–36] have shown that gender has no effect on financial literacy. In a limited number of studies [37], it is found that female students have a higher level of financial literacy than male students. The authors here noted that this may be possible since women have equal education opportunity and engage in financial topics.

According to Mottola's study [38], women tend to have lower income and financial literacy levels and they are less confident about their math skills and in as a consequence are more prone to engage into more costly credit card behaviours. To solve the problem of financial literacy, gender equity must be addressed as a central issue that goes beyond the economics of personal finance [39]. According to the authors, this issue should be considered as social justice.

Another factor influencing financial literacy is family and school. They play a key role in the education process, including the increase in financial knowledge. These are two basic environments where young people acquire knowledge in various areas among which finance. Family and school usually exert a crucial influence on the child's development [40–42]. They provide for experiences and knowledge and constitute one of the significant links in the chain of education. A crucial role of the family in financial education could be clearly visible in the results of the primary research, which demonstrated that most of the participants stated they received information mainly from family (30.3%), internet (9.4%), and school (7.1%). Surprisingly low is the share of school in raising financial literacy, which, however, can be quite easily explained with the fact that schools only rarely include financial issues in the curricula of subjects they teach. The role of the family is also clearly demonstrated by the fact that young people in need to solve a financial problem most frequently (74%) turn to their family for advice. University students in Portugal have low knowledge on banking issues, although they are aware of the importance of financial planning they do not invest but keep their money in checking accounts [43]. It was found that financial quiz scores of the university students in non-quantitative academic disciplines are higher than that of others [44]. According to another study, age may play a role in differentiating financial behaviour. In the older age groups, objective financial knowledge was more strongly related to long-term financial behaviour including retirement saving and investing behaviour. Knoll et al., [45] tried to create a new financial knowledge scale that will enable researchers to confidently compare financial knowledge across studies, populations, and time. They psychometrically created a 20-item financial knowledge scale known as the item response theory (IRT).

As we note from the above, in general, a comparison of the studies on financial literacy is a large problem, because of the different scales and methods used. In general most studies investigate financial literacy in university students, however, our aim is to understand financial literacy in Polish 15 year-old school students. Therefore, to do this we collect information on financial literacy and financial education in Poland, to reveal the gender effect on financial literacy and to understand the level of financial literacy of students in pre-undergraduate education and to compare the so called narrow studies of financial literacy to the so called broader studies noted above.

## 3. Methodology

This research was conducted between October 2017 and January 2018 and conducted among young people attending high school, with an average age of 15 years (i.e., meaning that they reached the age of 16 during January to December 2017, in line with the PISA survey methodology). The study was administered in schools during the first weeks of classes by their teachers. Therefore, there were students who had not yet attained the age of 15 years (1.9%) or were slightly older than 15 years (0.4%).

The survey was carried out in Poland on a nationwide representative sample of 2070 respondents. Due to some missing information, the final sample included 1932 respondents (82.8% response rate). They were the students of three types of schools: technical schools (17.6%), high schools (35.4%), and vocational schools (47%). The sample included also some younger or older subjects who started school a year earlier or were repeaters. A large majority (94%) of the surveyed students were born in 2001, 1.9% were born in 2000, and 0.4% in 2002, 3.7% of the subjects did not specify their date of birth, 52.7% of the participants were female, 46.6% male and 13 students (0.7%) did not mention their gender.

Table 1 shows the demographic characteristics of the respondents and displays some detailed information concerning their financial habits and the sources of their financial knowledge. Particularly, it demonstrates the role of family and school in this regard.

**Table 1.** Distribution and characteristics of the sample (N = 1932).

| Variable | Description | N | % |
|---|---|---|---|
| Date of birth | 2000 | 37 | 1.9 |
| | 2001 | 1816 | 94 |
| | 2002 | 8 | 0.4 |
| | Missing | 71 | 3.7 |
| | Total | 1932 | 100 |
| Gender | Male | 901 | 46.6 |
| | Female | 1018 | 52.7 |
| | Missing | 13 | 0.7 |
| | Total | 1932 | 100 |
| Parents education | **Mother** | | |
| | primary school | 259 | 13.4 |
| | trade/technical/vocational training | 613 | 31.7 |
| | high school graduate | 298 | 15.4 |
| | bachelor degree | 542 | 28.1 |
| | master degree/engineer | 57 | 3 |
| | **Father** | | |
| | primary school | 64 | 3.3 |
| | trade/technical/vocational training | 604 | 31.3 |
| | high school graduate | 566 | 29.3 |
| | bachelor degree | 191 | 9.9 |
| | master degree/engineer | 412 | 21.3 |
| Where do you find information about financial matters? | School | 138 | 7.1 |
| | Internet | 182 | 9.4 |
| | Family | 585 | 30.3 |
| | Family, Internet and School | 1013 | 52.4 |
| | Subtotal | 1918 | 99.3 |
| | Missing | 14 | 0.7 |
| | Total | 1932 | 100 |
| Are you getting regularly (pocket) money? | I do not get (pocket) money regularly | 762 | 39.4 |
| | From your mother/father/family | 920 | 47.6 |
| | I earn money by doing jobs | 208 | 10.8 |
| | Scholarship/Child allowance | 31 | 1.6 |
| | Subtotal | 1921 | 99.4 |
| | Missing | 11 | 0.6 |
| | Total | 1932 | 100 |

**Table 1.** *Cont.*

| Variable | Description | N | % |
|---|---|---|---|
| How well, in your opinion, do you manage your own money? | I have most of the time money problems | 549 | 28.4 |
| | I have to borrow money often | 858 | 44.4 |
| | only just as best as I can | 390 | 20.2 |
| | in a good way | 19 | 1 |
| | in a very good way | 110 | 5.7 |
| | Subtotal | 1926 | 99.7 |
| | Missing | 6 | 0.3 |
| | Total | 1932 | 100 |
| When in an average month you are running so short of money that you cannot anymore entertain something? | at the beginning of a month/after the first week | 130 | 6.7 |
| | midst of the month/after the second week | 407 | 21.1 |
| | at the end of the month/after the third week | 572 | 29.6 |
| | I have enough until the end of the month | 802 | 41.5 |
| | Subtotal | 1911 | 98.9 |
| | Missing | 21 | 1.1 |
| | Total | 1932 | 100 |
| Whom would you ask for advice if you had financial problems? | financial advisor, a teacher/school pedagogue, others | 101 | 5.2 |
| | Family and friend | 1429 | 74 |
| | nobody, would try to solve the problems by myself | 353 | 18.3 |
| | Subtotal | 1883 | 97.5 |
| | Missing | 49 | 2.5 |
| | Total | 1932 | 100 |
| In your opinion, the financial knowledge you get at home and school is enough to manage your own finances? | I strongly disagree | 120 | 6.2 |
| | I rather disagree | 477 | 24.7 |
| | I have no opinion | 503 | 26 |
| | I rather agree | 686 | 35.5 |
| | I strongly agree | 137 | 7.1 |
| | Subtotal | 1923 | 99.5 |
| | Missing | 9 | 0.5 |
| | Total | 1932 | 100 |
| What would you expect from a finance lesson at school? | theory/theoretical knowledge | 54 | 2.8 |
| | practical knowledge | 1505 | 77.9 |
| | practical and theory/theoretical knowledge | 278 | 14.4 |
| | I do not need special finance knowledge/ lessons | 90 | 4.7 |
| | Subtotal | 1927 | 99.7 |
| | Missing | 5 | 0.3 |
| | Total | 1932 | 100 |

Source: Authors' computation.

The study was completed using a Pencil and Paper Interview (PAPI) method. A test questionnaire was used, which contained questions concerning financial knowledge, financial skills, financial attitude, financial behaviour, and financial information obtained by young people.

Additional questions determined the demographic characteristics of the respondents. Financial knowledge was measured with true/false closed questions. With regards to financial attitude, respondents judged how much they agreed with various statements on a Likert-type scale. The part of the questionnaire concerning financial behaviour contained questions about savings, debt, consumption, and financial habits. Respondents choose from among several pre-existing answers. The total score could vary from 0 to 5 and the total amount scored was taken as the sum of the points obtained for each question.

In the area of financial knowledge, there were 12 true/false type questions. One point was assigned for a correct answer, '0' otherwise. Therefore, financial knowledge could assume a value from 0 to 12. Three levels of financial information were identified: low, average, and high, depending on the

number of correct answers. The low level was assigned to participants who answered correctly from 0 to 4 questions, the average one to those who gave from 5 to 8 correct answers and the high one to those who gave from 9 to 12 correct answers.

The reliability of the scale of financial knowledge was assessed. Bayram [46] stated that the reliability analysis measures the internal consistency of a set of items and gives information about the relationship between the items within a scale. In this study, the Cronbach's Alpha coefficient ($\alpha$) was calculated, a measure widely used to assess the reliability of questionnaires. The Cronbach Alpha ($\alpha$) reliability analysis results are shown in Table 2.

**Table 2.** Scale and reliability statistics.

| Scale Statistics | | | Reliability Statistics | |
|---|---|---|---|---|
| **Mean** | **Std. Deviation** | **N of Items** | **Cronbach's Alpha** | **N of Items** |
| 83.22 | 8.233 | 40 | 0.750 | 40 |

Source: Authors' computation.

The statistics displayed in Table 2 show that the general average is 83.22, the standard deviation for the scale is 8.233, and the reliability coefficient, Cronbach Alpha, is 0.75. Since the scale of reliability is between 0.60 and 0.80, it is considered reliable [47,48] and it can be said that the internal consistency of the data of this study is very good.

The primary survey was carried out in the fourth quarter of 2017. The questionnaires were sent to the teachers all over Poland, who distributed them to their students during the lessons and allowed the students an appropriate amount of time to complete them. In total 24 schools in 20 cities in 11 voivodeships participated in the survey.

## 4. Results

### 4.1. Financial Behaviour

Table 3 presents the results concerning financial behaviour, including savings, making debts, and consumption. The results show that more than 80% of young people are saving money. It should be noted, however, that this is a declarative answer, and that 30% are saving money irregularly, almost 40 per cent (39.8%) are saving for a specific target, and only about 11% (10.8%) are saving regularly. In our opinion saving money is a good financial behaviour, but at the age of 15 it doesn't predetermine anything. It is only natural that young people, having no financial liabilities and being provided for the necessities of life (board and lodging), save for a specific target (a new computer, a bicycle etc.). However, if a 15 year old is saving for a specific purpose or irregularly, it is not predetermined that he/she will continue this behaviour in the adult life. Of particular importance seems to be the result pointing to the regular saving of money, reported only by 10.8% of the respondents, as well as long-term saving, which was not addressed in this study, being more suitable for a study on the adults.

Above 80% (83.3%) of the respondents have no debts. However, it is slightly surprising that more than 13% (13.4%) owe money to their family. This result is high for young students, assuming the expectations that in the future they will be taking credit card consumer loans and subsequently, mortgages and car credits. Also, the young generation tend to incur debts, they grow up in comfort and wealth, are used to quality products and prices are secondary for them. They are often impatient, they want to have everything immediately and therefore take loans from banks and use credit cards. Although their main life task is learning and they are dependent on their parents, they are very active consumers and they willingly spend money on their whims. 42.2 per cent of the subjects consider themselves neither frugal nor extravagant, but as much as 40.6 per cent judge themselves as frugal (a rather frugal person 32.3% I am a very frugal person 8.3%).

**Table 3.** Financial behaviour.

| Question | Answer | Frequency | % |
|---|---|---|---|
| Are you regularly saving money? | No, I do not spare, I think I do not need the savings | 137 | 7.1 |
| | No, I do not save at all, I spend all my money | 214 | 11.1 |
| | Yes, but very irregular | 580 | 30.0 |
| | Yes. for a specific target (e.g., bicycle. computer etc.) | 768 | 39.8 |
| | Yes, I am saving regularly | 208 | 10.8 |
| What person do you consider yourself from the point of view of cash expenditures? | very spend thrift | 80 | 4.1 |
| | I'm rather spendthrift person | 245 | 12.7 |
| | neither frugal nor extravagant | 815 | 42.2 |
| | rather frugal person | 624 | 32.3 |
| | I am a very frugal person | 161 | 8.3 |
| Do you have at present any kind of debts? | I have debts for my mobile phone/for online shopping and others | 20 | 1.0 |
| | I owe money to my relatives (e.g., parents/brother/sister/grandparents) and to my friends | 259 | 13.4 |
| | No, I do not have debts | 1611 | 83.4 |
| | Subtotal | 1890 | 97.8 |
| | Missing | 42 | 2.2 |

Source: Authors' computation.

### 4.2. Financial Attitude

The results of the primary research show that the topic of money and talking about money is no taboo for the young Poles (Table 4). As many as 70% of the respondents claim that financial matters do not seem too complicated for them. They are generally not troubled by financial worries. However, despite the fact that because of their young age they perform rather simple financial operations, the same positive attitude to financial matters should be considered a good prognosis for the future. Of course, such an attitude may also result in reckless behaviour—It is simple, so I do not have to think more deeply about the financial aspects of the transaction. This approach may, with more complex operations such as credit, lead to imprudent financial decisions. On the basis of the conducted research, we cannot draw conclusions regarding such attitudes. However, the responsibility of adults (and of the educational system in the country) should—at the stage of education and upbringing—sensitize young people to financial matters and shape responsible attitudes. Young people should be aware that financial matters are of varying complexity and the more complex long-term ones require special attention and consideration when making financial decisions. However, young Poles declare awareness of the necessity of saving and as many as 76% of the respondents understand that refraining from current consumption and postponing (saving) money in this way may mean a better future for them. If we confront these attitudes with declarations regarding real behaviour (30% of the respondents declared that they regularly save and almost 40% save from time to time). These attitudes indicate high awareness of saving. Young Poles show quite reasonable attitudes about their spending options and declare that during the purchase of various products they wonder if they can afford such a purchase. Over 77% of the respondents declare that they do not buy things that they cannot afford. Only 17% of the respondents said that if they have money, they should not hesitate to flaunt it. The vast majority of respondents did not want to show in public that they have money. Such a result is interesting because it is widely believed that the young generation is very susceptible to the so-called consumer symbols of social status and thus – de facto—to flaunting the fact of having money. The research results show

that young people control their cash well, although the difference between the awareness of how much money is in their portfolio (84.5% of the respondents declare that they know) and how much money they have in their account (66% of the respondents declare that they know) is clearly visible. This difference may be due to the fact that many respondents do not have a bank account yet. In general, it should be stated that the respondents show great self-satisfaction in terms of their ability to deal with their financial affairs. Over 83% believe that they manage their money well [22].

**Table 4.** Financial attitudes.

| | | Freq. | % | | | Freq. | % |
|---|---|---|---|---|---|---|---|
| When we talk about money I get bad mood | Strongly Agree | 70 | 3.6 | If today I will withhold some expenses and save, I will be better off in the future | Strongly Agree | 400 | 20.7 |
| | Agree | 251 | 13.0 | | Agree | 1065 | 55.1 |
| | Disagree | 1116 | 57.8 | | Disagree | 383 | 19.8 |
| | Strongly Disagree | 485 | 25.1 | | Strongly Disagree | 76 | 3.9 |
| | Subtotal | 1922 | 99.5 | | Subtotal | 1924 | 99.6 |
| | Missing | 10 | 0.5 | | Missing | 8 | 0.4 |
| | Total | 1932 | 100 | | Total | 1932 | 100 |
| When I have savings I spend them soon to realize my wishes | Strongly Agree | 224 | 11.6 | I spend often more money than I intended too | Strongly Agree | 323 | 16.7 |
| | Agree | 634 | 32.8 | | Agree | 725 | 37.5 |
| | Disagree | 736 | 38.1 | | Disagree | 700 | 36.2 |
| | Strongly Disagree | 335 | 17.3 | | Strongly Disagree | 177 | 9.2 |
| | Subtotal | 1929 | 99.8 | | Subtotal | 1925 | 99.6 |
| | Missing | 3 | 0.2 | | Missing | 7 | 0.4 |
| | Total | 1932 | 100 | | Total | 1932 | 100 |
| I have the feeling that money and finances are too complicated for me | Strongly Agree | 98 | 5.1 | I often buy things without thinking whether I can afford them or not | Strongly Agree | 99 | 5.1 |
| | Agree | 477 | 24.7 | | Agree | 341 | 17.7 |
| | Disagree | 1031 | 53.4 | | Disagree | 975 | 50.5 |
| | Do not agree at all | 325 | 16.8 | | Do not agree at all | 513 | 26.6 |
| | Subtotal | 1931 | 99.9 | | Subtotal | 1928 | 99.8 |
| | Missing | 1 | 0.1 | | Missing | 4 | 0.2 |
| | Total | 1932 | 100 | | Total | 1932 | 100 |
| I attach a lot of importance to thrift | Strongly Agree | 653 | 33.8 | I know at the moment how much money I have got in my wallet/on my account | Strongly Agree | 886 | 45.9 |
| | Agree | 639 | 33.1 | | Agree | 746 | 38.6 |
| | Disagree | 357 | 18.5 | | Disagree | 216 | 11.2 |
| | Strongly Disagree | 257 | 13.3 | | Strongly Disagree | 81 | 4.2 |
| | Subtotal | 1906 | 98.7 | | Subtotal | 1929 | 99.8 |
| | Missing | 26 | 1.3 | | Missing | 3 | 0.2 |
| | Total | 1932 | 100 | | Total | 1932 | 100 |
| When I have money. I should not hesitate to flaunt it | Strongly Agree | 105 | 5.4 | I think I am capable of managing my money | Strongly Agree | 45 | 2.3 |
| | Agree | 218 | 11.3 | | Agree | 270 | 14.0 |
| | Disagree | 898 | 46.5 | | Disagree | 1209 | 62.6 |
| | Strongly Disagree | 698 | 36.1 | | Strongly Disagree | 402 | 20.8 |
| | Subtotal | 1919 | 99.3 | | Subtotal | 1926 | 99.7 |
| | Missing | 13 | 0.7 | | Missing | 6 | 0.3 |
| | Total | 1932 | 100 | | Total | 1932 | 100 |
| In my life everything will be somehow put in order | Strongly Agree | 554 | 28.7 | I am (actually) troubled by financial worries | Strongly Agree | 113 | 5.8 |
| | Agree | 970 | 50.2 | | Agree | 439 | 22.7 |
| | Disagree | 224 | 11.6 | | Disagree | 943 | 48.8 |
| | Strongly Disagree | 174 | 9.0 | | Strongly Disagree | 430 | 22.3 |
| | Subtotal | 1922 | 99.5 | | Subtotal | 1925 | 99.6 |
| | Missing | 10 | 0.5 | | Missing | 7 | 0.4 |
| | Total | 1932 | 100 | | Total | 1932 | 100 |

Source: Authors' computation.

*4.3. Financial Knowledge*

Last but not least, one of the most important elements of financial literacy is financial knowledge. In the primary research, 12 questions concerning the objective financial knowledge were used. The first question is about the balance sheet and is one of the most important financial statements. The balance sheet shows the assets and resources of an entity in a given period. Question 1 is about measuring the knowledge of young people about financial statements. The second question is about the budget and it is aimed to measure the knowledge of young people about financial planning. The third question is about debit card and it is aimed to measure the basic banking knowledge of young people. The fourth question is about retirement and savings and it is aimed at researching the savings and retirement information of young people. Question 5 is about banking and it is aimed to measure the knowledge of young people about bank account. Question 6 is about financial planning and it is aimed to measure the knowledge of young people about planning. Question 7 is aimed at measuring whether young people know who can benefit from financial information. Question 8 is about banking and it is tried to learn the Overdraft Account information of young people. Question 9 is about measuring the knowledge of young people about credit card. The 10th question is about the online banking question that almost everyone has started to use recently and has tried to measure the knowledge of young people on this issue. Question 11 is aimed at measuring mobile banking knowledge of young people. The 12th question is about digital money. Bitcoin is one of the most well-known digital coins. It is created and maintained electronically. It is the first example of a growing currency category known as cryptocurrency. With this question, we investigated how much this currency, which is foreseen to be widely used in the future, is known by young people.

These information are not an opinion of one's level of knowledge, but the real knowledge of some issues concerning personal finances to which participants were required to answer true/false question. The analysis of financial knowledge demonstrated that there were much more correct answers than incorrect ones. The number of all the correct answers exceeded 50% and in some subjects, this even reached 90%. The incorrect answers comprised both false answers and the answers "I don't know" (Table 5). Both these types of incorrect answers demonstrated that the respondents had no knowledge on a given subject. It is worth noting that the number of answers "I don't know" is quite considerable—from 9.1% to the question "if one can pay for goods and services by mobile phone" to 25,9% to the question on the debit and credit payment cards to 38.3% to the question concerning bitcoin. Young people have some basic knowledge of personal finances, but when it comes to the more advanced services and instruments, they need further education. The results confirm the necessity to elaborate on the subject and provide them with proper education. It should be emphasized that possessing knowledge not always translates into proper financial behaviour and attitudes towards money and therefore, besides financial knowledge, a significant role should be assigned to emotions, environment in which the youth is brought up, their own experiences and those of their parents and family, as well as the place where they live in.

At first sight, it is clear from Table 5 that the students knew the best correct answer to the eleventh question. On the other hand, it is also clear that the largest number (94.5%) of wrong answers was given to the first question. Apart from this, half of the students solved correctly eleven of the twelve problems.

**Table 5.** Financial Knowledge.

| Knowledge Statements | Correct Answer (%) | Incorrect Answer (%) | Do Not Know (%) |
|---|---|---|---|
| Balance sheet is a statement of costs and revenues for a given period. | 5.5 | 73.5 | 21.0 |
| Budget is a financial plan containing a statement of income and expenditure. | 59.6 | 25.8 | 14.6 |
| The debit card works just like a credit card. | 63.8 | 10.3 | 25.9 |
| Income levels affect future retirement pension. | 78.0 | 9.6 | 12.4 |
| Bank accounts may only be owned by an adult. | 80.7 | 9.8 | 9.5 |
| Financial planning is finding the best ways to use your money. | 68.6 | 11.4 | 20.0 |
| Financial knowledge is only needed for those who work in financial institutions. | 77.3 | 8.6 | 14.1 |
| Overdraft is the amount that an account holder can debit at a given bank. | 59.0 | 11.0 | 30.0 |
| Visa/MasterCard are only for withdrawing cash from an ATM. | 75.2 | 9.0 | 15.8 |
| Only adults can use the online banking services. | 57.0 | 20.0 | 23.0 |
| You can pay for goods and services by mobile phone. | 85.6 | 5.3 | 9.1 |
| Bitcoin is money you can use to pay online. | 55.5 | 6.2 | 38.3 |

Source: Authors' computation.

Table 6 highlights the score levels of financial literacy obtained from the survey.

**Table 6.** Levels of financial knowledge.

| Levels | Level of Financial Knowledge | Number of Correct Answers | N | % |
|---|---|---|---|---|
| Level 1 | Low level | 0–4 | 209 | 10.8 |
| Level 2 | Average level | 5–8 | 876 | 45.3 |
| Level 3 | High level | 9–12 | 847 | 43.8 |
| | TOTAL | | 1932 | 100 |

Source: Authors' computation.

Accordingly, 209 students answered correctly up to 4 questions, 876 students answered from 5 to 8 questions correctly and 847 students answered from 9 to 12 questions correctly. The largest number of students were scored at level 2, although the number of students scoring at level 3 is very close to those who scored level 2. Therefore, one can say that the majority of students reached a level score, which was between average and high (Table 6).

Table 7 presents more detailed information on the scores achieved by the participants according to gender.

According to the results of the primary research, financial knowledge of the female and male participants is similar, although on average women scored higher (64%) than men (63.7%). Women proved to have better financial knowledge on the issues asked in 9 out of 12 questions, while men were better informed only on 3 of them. That is balance sheets, payment cards and bitcoins.

As highlighted in Table 7, the overall score of students was 63.8%. Students gave the best answers to question 11 (85.6%) and the poorest answers to question 1 (5.5%). Therefore, it can be said that there are little differences between genders in terms of the financial knowledge level, although in general, men or women are better informed on particular subjects.

**Table 7.** Average financial knowledge scores according to gender.

| Questions | Gender | | General Scores |
|---|---|---|---|
| | Female | Male | |
| 1. Balance sheet is a statement of costs and revenues for a given period | 4.5% (46/1018) | 6.5% (59/901) | 5.5% (105/1919) |
| 2. Budget is a financial plan containing a statement of revenue and expenditure | 62.9% (640/1018) | 55.9% (504/901) | 59.6% (1144/1919) |
| 3. The debit card works just like a credit card | 64.1% (653/1018) | 63.5% (572/901) | 63.8% (1225/1919) |
| 4. Income levels affect future retirement pension | 79.2% (806/1018) | 76.7% (691/901) | 78.0% (1497/1919) |
| 5. Bank accounts may only be owned by an adult | 81.7% (832/1018) | 79.5% (716/901) | 80.7% (1548/1919) |
| 6. Financial planning is finding the best ways to use your money | 70.5% (718/1018) | 66.4% (598/901) | 68.6% (1316/1919) |
| 7. Financial knowledge is only needed for those who work in financial institutions | 79.5% (809/1018) | 74.8% (674/901) | 77.3% (1483/1919) |
| 8. Overdraft is the amount that an account holder can debit at a given bank | 61.9% (630/1018) | 55.8% (503/901) | 59.0% (1133/1919) |
| 9. Visa/MasterCard are only for withdrawing cash at an ATM | 74.7% (760/1018) | 75.9% (684/901) | 75.2% (1444/1919) |
| 10. Only adults can use the online banking services | 58.5% (596/1018) | 55.3% (498/901) | 57.0% (1094/1919) |
| 11. You can pay for goods and services by mobile phone | 87.7% (893/1018) | 83.1% (749/901) | 85.6% (1642/1919) |
| 12. Bitcoin is money you can use to pay online | 42.3% (431/1018) | 70.5% (635/901) | 55.5% (1066/1919) |
| Mean | 64.0% | 63.7% | 63.8% |

Source: Authors' computation.

The independent t-tests were used to determine whether there is a significant difference between gender and the levels of financial knowledge, financial behaviours and the ability to use financial instruments. Before conducting these tests, the normality of the distribution of the data was assessed. To this end, the skewness and kurtosis were evaluated. The values of the skewness and kurtosis are presented in Table 8.

**Table 8.** Descriptive statistics.

| | Financial Knowledge | Financial Behaviour | Financial Attitudes |
|---|---|---|---|
| **N** | 1932 | 1932 | 1931 |
| **Mean** | 0.6399 | 2.9233 | 2.9161 |
| **Median** | 0.6667 | 2.9091 | 2.9982 |
| **Skewness** | −1.091 | −0.920 | −0.138 |
| **Kurtosis** | −0.095 | 0.177 | 0.040 |

Source: Authors' computation.

Results show that the skewness values are −1.091 for financial knowledge, −0.92 for financial behaviour and −1.138 for financial attitudes. The kurtosis values are −0.095 for financial knowledge, 0.177 for financial behaviour and 0.040 for financial attitudes. Since all the skewness and kurtosis values are between +1.50 and −1.50, it can be said that the data is normally distributed [49].

The results of the t test for financial knowledge are shown in Table 9.

**Table 9.** The results of the t test for financial knowledge scores according to gender.

| Gender | N | Mean | Std. Deviation | F | P |
|--------|-----|--------|----------------|--------|-------|
| Female | 1018 | 0.6397 | 0.19044 | 16.620 | 0.746 |
| Male | 901 | 0.6366 | 0.22065 | | |

Source: Authors' computation.

Results show that the average financial knowledge score of 1018 female students is 0.6397 and that of 901 male students is 0.6366. However, although the achievement level of female students is slightly higher than that of male students, the difference is not significant ($P = 0.746 > 0.05$).

The results of the t-test for financial behaviour are presented in Table 10.

**Table 10.** The results of the t-test for financial behaviour according to gender.

| Gender | N | Mean | Std. Deviation | F | P |
|--------|-----|--------|----------------|-------|-------|
| Female | 1018 | 2.8722 | 0.38602 | 2.603 | 0.000 |
| Male | 900 | 2.9841 | 0.36804 | | |

Source: Authors' computation.

In Table 10 we show the results that the average financial behaviour score of 1018 female students is 2.8722 and that of 900 male students is 2.9841 and that this difference is statistically significant ($P = 0.000 < 0.05$).

The results of the t-test for the use of financial skills are presented in Table 11.

**Table 11.** The results of the t-test for financial skills according to gender.

| Gender | N | Mean | Std. Deviation | F | P |
|--------|-----|--------|----------------|-------|-------|
| Female | 1015 | 1.7189 | 0.29989 | 6.517 | 0.000 |
| Male | 888 | 1.8026 | 0.27596 | | |

Source: Authors' computation.

Accordingly, results show that the average financial skills for 1015 female students is 1.7189 and that of 888 male students is 1.8026 and this difference is statistically significant ($P = 0.000 < 0.05$).

## 5. Discussions and Conclusions

Increasing financial literacy plays an important role in achieving sustainable economic development and individual welfare. Therefore, the financial literacy level of the society should be investigated. The level of financial knowledge of individuals, the ability to use financial instruments and methods correctly should be determined. Appropriate training programs can be designed as a result of diagnosing individuals' financial deficiencies.

With this study our objective was to understand and determine the level of financial literacy of high school students in Poland and to determine whether financial literacy changes with gender. From the literature above, one notes that research in financial literacy mostly conducted on households [26–30] and university students [31–33]. As already noted the most important element that distinguishes our study from the others is that our study was carried out with a large sample of high school students with an average age of 15–16 years. In addition, we analysed the effect of gender on financial literacy at this early age and compared the wider themes to the so-called narrow themes.

The results of the research demonstrated a good and partially very good, level of financial knowledge in young Polish students. 45.3% obtained an average level score and 43.8% achieved a high-level score in financial knowledge. This result explains the rationality of students in their financial decisions. However, as noted through the literature above, this cannot be seen in isolation as knowledge is only one of the components required to ensure future financial livelihood and economic growth.

Also, in this study we found that results on gender differences on the level of financial knowledge are insignificant. That is, there is no difference whether one is male or female. However, results show that gender makes a difference on financial behaviour and financial skills and the financial literacy level of males is higher than females.

Although, as noted above there are a few studies that do not concur to our results [36], they corroborate to most of the findings of studies on households [14,26–30] and on university students [30,32,33]. Moreover, our findings also show that males have higher financial literacy levels than females in younger age groups. Another important result of the study is that family members occupy a very important place as a source of financial knowledge for high school students.

One should note, that although, undoubtedly, knowledge lays a necessary foundation, a high level of financial knowledge not always translates into the proper financial behaviour on financial markets. This can be explained using an analogy of commonly used diets - we might know a lot about diets, but it doesn't necessarily mean that we can apply this knowledge and achieve the desired results. When supplemented with the experience—one's own and that of the parents, family and the closest environment; it should bring positive results in financial life and livelihood. This, however, is just the base that should be developed so as to know how to plan a budget, to save, to invest or to know what loans to take and when, how to manage debt, how to protect oneself for the future, etc.

The appropriate way to acquire financial knowledge that will translate into proper behaviour is financial education. Financial education should be carefully thought of and well adapted to the age of a student. It shouldn't be uniform for all since the needs of young people are different from those of adults. One must also consider the environment, culture and background which can differ between regions and states. The language and terminology should be adequately adapted so that the message could reach its recipient. Young students, today are in many ways privileged since they have access to the world wide web of the internet, online books, articles, blogs, and social media, etc. where they can find a lot of information on financial matters, which of course might not have been available to their parents and grandparents. Financial literacy is of great importance not only for the individuals but also for the state economy. The more financial awareness individuals have, the more they expect from the financial institutions, the authorities and the employers.

The findings of the study provide evidence that national and international organizations should continue to give importance to financial literacy in females by enhancing programs for all ages. Therefore, it is suggested that further studies should be conducted to determine the effectiveness of special and common education programs. Financial literacy also affects countries' education curriculum. For example, the Ministry of Education of the Republic of Turkey has decided to add 2 courses (behavioural economics and financial mathematics) as elective courses to the high school curriculum. Also, it would be ideal if further studies are carried out on the different financial literacy education projects and programs provided by different countries, using these as laboratory tests to determine their success to translate into a stronger economy and whether these can be used in Poland [50,51].

**Author Contributions:** Conceptualization, B.S., E.Y., E.Ö. and S.G.; methodology, B.S., E.Y., E.Ö. and S.G.; software, E.Y. and E.Ö.; validation, B.S., E.Y., E.Ö. and S.G.; formal analysis, B.S., E.Y., E.Ö. and S.G.; investigation, B.S., E.Y., E.Ö. and S.G.; resources, B.S., E.Y., E.Ö. and S.G.; data curation, B.S., E.Y., E.Ö. and S.G.; writing—original draft preparation, B.S., E.Y., E.Ö. and S.G.; writing—review and editing, B.S., E.Y., E.Ö. and S.G.; visualization, B.S., E.Y., E.Ö. and S.G.; supervision, B.S., E.Ö. and S.G.; project administration, S.G.; funding acquisition, B.S. All authors have read and agreed to the published version of the manuscript.

**Funding:** The project is financed within the framework of the program of the Minister of Science and Higher Education under the name "Regional Excellence Initiative" in the years 2019–2022; project number 001/RID/2018/19; the amount of financing PLN 10,684,000.00.

**Acknowledgments:** Before restructuring, this paper was presented during the 22nd Finance Symposium in Mersin, Turkey, on 12–15 October 2018.

**Conflicts of Interest:** The authors declare no conflict of interest.

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
