# Peer review of "Financial Literacy: The Case of Poland"

_sustainability, doi:10.3390/su12020700_

Round 1

Reviewer 1 Report

The (ambitious) aim of the paper is to: 1. offer new insights in the definition (and conceptualization) of financial literacy (FL). 2. compare recent results of a survey on the FL of 15-years old Polish, comparing these results with Pisa results obtained by Polish teen agers in 2015 and 2012 surveys

Nothing of what promised is delivered; the paper is simply, in the present format, a description of the results of a survey recently made (Oct 2017-Jan 2018).

In what follows, the authors can find some suggestion on how to improve their paper 

First of all, make sure that all authors quoted in the text are present in the reference section!!! Starting from Bryant (2013) in the second line of the first statement. Too many are missing. Make sure that the paper whose methodology you say you follow in the construction of your survey is properly cited. (Robson (2012))

Second, the conceptualization of financial literacy is a serious thing. I had many expectations when decided to accept the revision of the paper. I did it because you mentioned that you were trying to add - even with a minor idea - to this stream of literature. What a delusion. You are just (too briefly) summarizing previous literature, not even with a critical view and then conclude that you follow Robson (2012) (not present in the reference section) according to which FL has 3 dimensions : knowledge, attitude and behavior ...the ones that by the way inform the OECD/INFE approach...

Houston (2010) and Remund (2010) are excellent literature reviews helping to frame the issue of the conceptual definition of what financial literacy is or should be. Also Knoll and Houts (2012) should be cited

If you do not have much to say (that was already highlighted)  about the problems of definition of a latent variable as FL is, and the types of questions to be used to measure such an un-observable variable, then drop this part

Third, your survey on Polish 15 years old ....who in fact are 16 years old (if the majority of the sample respondents were born in 2001 and you started your survey in Oct 2017!) Make sure your statements are correct with respect to the data provided and commented

You should explain why you need to do a new survey given that PISA survey are done with regularity...

One major motivation is that you do not believe in the definition, conceptualization and measurement made by the PISA consortium and want to propose a different set of questions 

but then you should devote your study on this comparison, criticizing the PISa construct (and not limiting to describe how badly Polish youngsters performed)

then you should prove that your construct is much better in measuring the level of financial literacy. For instance why knowing the definition of a balance sheet is a good measure for FK?

Once this is done, your paper is ready for immediate publication

there is a huge need to challenge the present constructs used to define FL

Author Response

Reviewer 1

The (ambitious) aim of the paper is to: 1. offer new insights in the definition (and conceptualization) of financial literacy (FL). 2. Compare recent results of a survey on the FL of 15-years old Polish, comparing these results with Pisa results obtained by Polish teen agers in 2015 and 2012 surveys

Nothing of what promised is delivered; the paper is simply, in the present format, a description of the results of a survey recently made (Oct 2017-Jan 2018).

In what follows, the authors can find some suggestion on how to improve their paper 

First of all, make sure that all authors quoted in the text are present in the reference section!!! Starting from Bryant (2013) in the second line of the first statement. Too many are missing. Make sure that the paper whose methodology you say you follow in the construction of your survey is properly cited. (Robson (2012))

Thank you for your valid comments and please note that all references have been checked and included.

Reference [1] Bryant, J.H. Economic Growth and Sustainability Rooted in Financial Literacy. In: Madhavan G., Oakley B., Green D., Koon D., Low P. (eds) Practicing Sustainability. Springer, New York, NY.  2013 https://doi.org/10.1007/978-1-4614-4349-0_19.

Reference [20] [24] Robson, J. The case for financial literacy: Assessing the effects of financial literacy interventions for low income and vulnerable groups in Canada. Ottawa, ON: SEDI Canadian Centre for Financial Literacy. 2012.

Please change the referencing as per journal requirement and check that all are in text and reference list

Agreed and Done

Second, the conceptualization of financial literacy is a serious thing. I had many expectations when decided to accept the revision of the paper. I did it because you mentioned that you were trying to add - even with a minor idea - to this stream of literature. What a delusion. You are just (too briefly) summarizing previous literature, not even with a critical view and then conclude that you follow Robson (2012) (not present in the reference section) according to which FL has 3 dimensions : knowledge, attitude and behavior ...the ones that by the way inform the OECD/INFE approach...

Thank you for your comments which have added value to our article.

We have as suggested included the Reference to [20] [24] Robson, J. The case for financial literacy: Assessing the effects of financial literacy interventions for low income and vulnerable groups in Canada. Ottawa, ON: SEDI Canadian Centre for Financial Literacy. 2012.

Moreover, we have adjusted our paper to show the significance and the distinguishing aim of our article in that with the study we aim to determine the level of financial literacy of high school students in Poland and to determine whether financial literacy changes according to gender. The most important element that distinguishes our study from the others is that our study was carried out with a large sample of high school students with an average age of 15-16 years. In addition, the effect of gender on financial literacy at an early age was investigated, also comparing the wider themes to the so called narrow themes. (Line 33 to 72 and 189 to 202)

We have adjusted the results section and the conclusion to show that there is a good and partially very good, level of financial knowledge of the young people in Poland. 45.3 % obtained an average level score and 43.8 % achieved a high-level score in financial knowledge and that they can be rational in their financial decision making. However we note that although, it is understood that gender makes a difference on financial behaviour and use of financial instruments, gender does not make any difference on the level of financial knowledge. Moreover, we note that the financial literacy level of males is found to be higher than females. We discuss this in line with literature.

Huston (2010) and Remund (2010) are excellent literature reviews helping to frame the issue of the conceptual definition of what financial literacy is or should be. Also Knoll and Houts (2012) should be cited

Thank you for these valid suggestions which have been included as follows: Knoll et all was cited in line 178 ref. (47), Remund cited in line 100 ref. (18) and Huston cited in line 77 ref. (6)

If you do not have much to say (that was already highlighted)  about the problems of definition of a latent variable as FL (Financial Literacy) is, and the types of questions to be used to measure such an unobservable variable, then drop this part

The conceptual part has been expanded to include a broader explanation, indicating at the same time the contribution and its own approach to the financial literacy problem, as noted above.

Third, your survey on Polish 15 years old ....who in fact are 16 years old (if the majority of the sample respondents were born in 2001 and you started your survey in Oct 2017!) Make sure your statements are correct with respect to the data provided and commented

Thank you for this comment which has helped us clarify this point in Line 190 to 200, which explains that this research was conducted between October 2017 and January 2018 and conducted among young people attending high school, with an average age of 15 years (i.e. meaning that they reached the age of 16 during January to December 2017, in line with the PISA survey methodology). The study was administered in schools during the first weeks of classes by their teachers. Therefore, there were students who had not yet attained the age of 15 years (1.9%) or were slightly older than 15 years (0.4%). The survey was carried out in Poland on a nationwide representative sample of 2070 respondents. Due to some missing information, the final sample included 1932 respondents (82.8% response rate). They were the students of three types of schools: technical schools (17.6%), high schools (35.4%), and vocational schools (47%). The sample included also some younger or older subjects who started school a year earlier or were repeaters. A large majority (94 %) of the surveyed students were born in 2001, 1.9 % were born in 2000, and 0.4 % in 2002,

Moreover, although it may sound rather strange, this is in line with the PISA study those who obtained the age of 16 during 2017 were considered as 15 years of age for this study since during October 2017, they might have not yet obtained the age of 16.

You should explain why you need to do a new survey given that PISA survey are done with regularity...

Thank you for this suggestion which has helped to clarify the need for a new study, please note that while there is no doubt about the scope, value and validity of the PISA research, mainly due to its international character and its rich source of information on financial literacy, however, we believe that financial literacy can be supplemented with aspects included in the research presented in this article. The research presented in the article does not have the impetus of PISA, but complements it by indicating aspects not tested or tested using different methods from the PISA survey. Example using the narrow approach, the different timing and putting some more emphasis on the gender aspect.

One major motivation is that you do not believe in the definition, conceptualization and measurement made by the PISA consortium and want to propose a different set of questions. but then you should devote your study on this comparison, criticizing the PISa construct (and not limiting to describe how badly Polish youngsters performed). then you should prove that your construct is much better in measuring the level of financial literacy. For instance why knowing the definition of a balance sheet is a good measure for FK (Financial Knowledge)?

Thank you for this suggestion which has helped to make our article better. Our Primary research covered the elements of finance that young people are most interested in. It covered those aspects of finance that are adapted to the age of the researcher, including home budget, balance sheet, payment card payments and knowledge on how to set up a bank account. These were not mentioned in some other studies.

The advantage of this research is that it distinguishes between levels of financial knowledge, (i.e. low, medium and high) and looks at objective financial knowledge which is more reliable Lines 308 to 311 and Lines 352 to 355.

Financial statements are among the basic information to be learned and used for decision making. Knowledge of the make-up of financial statements helps in the analyses of companies with regards to efficiency and profitability. Thus, it helps individuals to invest more accurately in capital markets. Which as noted in the text leads the way to have more production, resulting in more income, employment and wealth. Line 57 to 61

Reviewer 2 Report

The authors of the research paper Financial Literacy: The Case of Poland presents a relevant topic considering the current context of the ever-changing banking financial services, which confirms the important role of financial literacy at the level of individuals, families, enterprises and national economies.
The bibliographic concepts and sources used by the authors of the research are adequate, demonstrating that "financial literacy is a significant element in understanding finance and rational financial decision-making" and directly influences the financial life quality of users of financial sources. Moreover, the authors justify that financial literacy leads to rational decisions in the field of finance.
The research methodology is relevant to the research, the authors of the research using classical research instruments, respectively the international survey on young people (15-year students - the PISA program) based on a test questionnaire. Moreover, the authors of the research use the OECD methodology (2011), defining "financial literacy as a combination of financial behavior, financial attitude and financial knowledge", as well as the use of the PAPI method.

Results are presented adequately by the research authors based on the research instrument questionnaire, respectively the research authors present the interpreted values ​​of the answers of the interviewees based on the questionnaire, the kurtosis values ​​being -0.095 for financial knowledge, 0.177 for financial behavior and 0.040 for financial attitudes. However, we suggest the authors of the research to present the personal contribution of the research to the thematic researched from the scientific point of view, as well as the pragmatic aspects resulting from the research.

The conclusions are appropriate, the authors of the research demonstrating that financial literacy is of particular importance not only to individuals, but also to the state economy, but we suggest the authors of the research to present the conclusions both from the point of view of the authors' contribution to the scientific bibliography, but also to the literature the utility of research results for pragmatic financial literacy at the individual and society level.

Therefore, after the review by the research authors of the chapter presenting the results and conclusions, we propose the paper is well-written, financial literacy being especially important for the young generation subjected to multiple challenges in the financial field.

Author Response

The authors of the research paper Financial Literacy: The Case of Poland presents a relevant topic considering the current context of the ever-changing banking financial services, which confirms the important role of financial literacy at the level of individuals, families, enterprises and national economies. The bibliographic concepts and sources used by the authors of the research are adequate, demonstrating that "financial literacy is a significant element in understanding finance and rational financial decision-making" and directly influences the financial life quality of users of financial sources. Moreover, the authors justify that financial literacy leads to rational decisions in the field of finance.
The research methodology is relevant to the research, the authors of the research using classical research instruments, respectively the international survey on young people (15-year students - the PISA program) based on a test questionnaire. Moreover, the authors of the research use the OECD methodology (2011), defining "financial literacy as a combination of financial behavior, financial attitude and financial knowledge", as well as the use of the PAPI method.

Results are presented adequately by the research authors based on the research instrument questionnaire, respectively the research authors present the interpreted values ​​of the answers of the interviewees based on the questionnaire, the kurtosis values ​​being -0.095 for financial knowledge, 0.177 for financial behavior and 0.040 for financial attitudes.

However, we suggest the authors of the research to present the personal contribution of the research to the thematic researched from the scientific point of view, as well as the pragmatic aspects resulting from the research.

We thank you for your suggestion which has helped us adjust our article to show the significance and our distinguishing aim, in that with the study we aim to determine the level of financial literacy of high school students in Poland and to determine whether financial literacy changes according to gender. The most important element that distinguishes our study from the others is that our study was carried out with a large sample of high school students with an average age of 15-16 years. In addition, the effect of gender on financial literacy at an early age was investigated, also comparing the wider themes to the so-called narrow themes. (Line 33 to 72 and 189 to 202)

We have adjusted the results section and the conclusion to show that there is a good and partially very good, level of financial knowledge of the young people in Poland. 45.3 % obtained an average level score and 43.8 % achieved a high-level score in financial knowledge and that they can be rational in their financial decision making. However, we note that although it is understood that gender makes a difference in financial behaviour and use of financial instruments, gender does not make any difference in the level of financial knowledge. Moreover, we note that the financial literacy level of males is found to be higher than females. We discuss this in line with the literature.

The conclusions are appropriate, the authors of the research demonstrating that financial literacy is of particular importance not only to individuals, but also to the state economy, but we suggest the authors of the research to present the conclusions both from the point of view of the authors' contribution to the scientific bibliography, but also to the literature the utility of research results for pragmatic financial literacy at the individual and society level.

We thank you for your suggestion which has helped us better our article conclusions and introduction where we note the need, importance and significance of carrying out such a study and make recommendations for further studies on the subject. Line 34 to 42 and 410 to 463

Reviewer 3 Report

The paper addresses interesting and important topic of the financial literacy of young people, using the example of young people in Poland. The study is based on sound methodology and the results are interpreted and discussed correctly. There are some rather minor issues that could be changed in order to improve the paper even further.

The aims of the study presented throughout the paper are inconsistent and, even more importantly, some of them do not fully reflect the discussed issues. The structure of the paper should be improved. For example, in Section 2 (entitled 'Methods...') discussion focuses on PISA and factors that affect financial literacy - those two issues are weakly linked and not related to the title of the Section. It should rather concentrate on the measures of financial literacy and show various approaches. Methodological section is rather confusing as it includes to a large extent the presentation of results rather than the outline of research methods. These two issues should be clearly separated, Authors should give more details about the differences between the various types of schools that were surveyed, e.g., technical vs vocational school. Some of the answers that were used in the survey seem to some degree overlapping. In my opinion repeating certain parts of Table 1 as Tables 3 etc. is not necessary and they are redundant - I suggest providing the names of subcategories within Table 1. There should be more in-depth explanation why certain statements were examined as survey of financial knowledge. What was their purpose - what types of knowledge were checked? There are some apparently conflicting results in Tables 5 and 7 - see, e.g., the scores for the first statement. The last Section requires significant extensions. The results of the analysis should be compared to previous studies. Limitations of the analysis as well as directions for the future studies should be provided. References are inconsistent with the journal's requirements.

Author Response

Reviewer 3

The paper addresses interesting and important topic of the financial literacy of young people, using the example of young people in Poland. The study is based on sound methodology and the results are interpreted and discussed correctly. There are some rather minor issues that could be changed in order to improve the paper even further.

The aims of the study presented throughout the paper are inconsistent and, even more importantly, some of them do not fully reflect the discussed issues. The structure of the paper should be improved. For example, in Section 2 (entitled 'Methods...') discussion focuses on PISA and factors that affect financial literacy - those two issues are weakly linked and not related to the title of the Section. It should rather concentrate on the measures of financial literacy and show various approaches.

           We thank you for your suggestion which has helped us adjust the structure so that there is a clearer triangulation between the sections, mainly the aim, the significance, the methodology (Line 190 to 220), the literature, the findings and the conclusions. With the study, we aim to determine the level of financial literacy of high school students in Poland and to determine whether financial literacy changes according to gender. The most important element that distinguishes our study from the others is that our study was carried out with a large sample of high school students with an average age of 15-16 years. In addition, the effect of gender on financial literacy at an early age was investigated, also comparing the wider themes to the so-called narrow themes. (Line 33 to 72 and 189 to 202).

We have adjusted the results section and the conclusion to show that there is a good and partially very good, level of financial knowledge of the young people in Poland. 45.3 % obtained an average level score and 43.8 % achieved a high-level score in financial knowledge and that they can be rational in their financial decision making. However, we note that although it is understood that gender makes a difference in financial behaviour and use of financial instruments, gender does not make any difference in the level of financial knowledge. Moreover, we note that the financial literacy level of males is found to be higher than females. We discuss this in line with the literature.

Methodological section is rather confusing as it includes to a large extent the presentation of results rather than the outline of research methods. These two issues should be clearly separated, Authors should give more details about the differences between the various types of schools that were surveyed, e.g., technical vs vocational school. Some of the answers that were used in the survey seem to some degree overlapping. In my opinion repeating certain parts of Table 1 as Tables 3 etc. is not necessary and they are redundant - I suggest providing the names of subcategories within Table 1.

We thank you for your suggestion to distinguish between the responses by students from different schools, which although we agree can help make our paper more interesting, was beyond the scope of this paper and the information had not collected. However, Table 1 and 3 were amended and arranged as suggested

There should be more in-depth explanation why certain statements were examined as survey of financial knowledge. What was their purpose - what types of knowledge were checked? There are some apparently conflicting results in Tables 5 and 7 - see, e.g., the scores for the first statement.

We thank you for your suggestion which has helped us adjust the section on financial Knowledge to include an explanation of the statements (Line 306 to 324).

Moreover, Table 5 and 7 were amended and arranged as suggested to make them consistent